# ^1^H-NMR Approach for the Discrimination of PDO Grana Padano Cheese from Non-PDO Cheeses

**DOI:** 10.3390/foods13030358

**Published:** 2024-01-23

**Authors:** Valentina Maestrello, Pavel Solovyev, Pietro Franceschi, Angelo Stroppa, Luana Bontempo

**Affiliations:** 1Fondazione Edmund Mach (FEM), Centre for Research and Innovation (CRI), Via E. Mach 1, 38098 San Michele all’Adige, TN, Italy; valentina.maestrello@unitn.it (V.M.); pietro.franceschi@fmach.it (P.F.); luana.bontempo@fmach.it (L.B.); 2Centre for Agriculture, Food and Environment (C3A), University of Trento, Via E. Mach 1, 38098 San Michele all’Adige, TN, Italy; 3Consorzio Tutela Grana Padano, Via XXIV Giugno 8, San Martino Della Battaglia, 25010 Desenzano del Garda, BS, Italy; a.stroppa@granapadano.com

**Keywords:** NMR spectroscopy, metabolomics, PDO cheese, authenticity

## Abstract

Protected Designation of Origin cheeses are products with high-quality standards that can claim higher prices on the market. For this reason, non-PDO cheeses with lower quality can be mislabeled as PDO or mixed with it for economic gain especially when the product is in a shredded form. Luckily, the production of PDO cheese is subjected to strict procedural specification rules that result in a product with a defined profile of its metabolites, which can be used for authentication purposes. In this study, an NMR metabolomic approach combined with multivariate analysis was implemented to build a classification model able to discriminate PDO Grana Padano cheese from a large dataset of competitors. The great advantage of the proposed approach is a simple sample preparation, obtaining a holistic overview of the analyzed samples. The untargeted approach highlighted a “typical profile” of Grana Padano samples, which could be used for protection purposes. In parallel, the targeted results allowed us to identify potential chemicals, such as lactate, some amino acids and lipids. These initial results could open the road to a potential new additional tool to check the authenticity of PDO cheeses in the future.

## 1. Introduction

In recent years, there has been an increase in the number of food frauds, among which the mislabeling of Geographical Indication is particularly frequent. As reported by Coldiretti, Grana Padano cheese is in second position in the ranking of the most counterfeited “made in Italy” products [1], even though the European Regulation is working to protect it. European law (EU) No 1151/2012 protects PDO (Protected Designation of Origin), PGI (Protected Geographical Indication) and TSG (Traditional Specialty Guarantee) food products against mislabeling [2,3], but this is not enough, and imitators keep finding new ways to mislead consumers. The 2021 Report by the Italian Inspectorate for fraud repression and quality protection of agri-food products and foodstuffs (ICQRF) also mentioned 55 interventions for the protection of PDO Grana Padano cheese in the period 2015–2021, found both on the web and ex officio, showing that it is a food product at the very center of imitation issues [4]. The price of Grana Padano has increased in the last few years, and it is one of the most expensive cheeses in the world. This makes it the ideal target for frauds, in which lower-quality non-PDO cheeses are labeled as PDO Grana Padano cheese or mixed with it for economic gain [5]. Even if the high quality of Grana Padano is protected by strict procedural specification rules for its production [6], especially when the cheese is grated or shredded, the mislabeling is much easier, because the “quality certification” mark printed on the crust is missing.

The analytical method already approved by the European law to verify the authenticity and traceability of Grana Padano cheese is based on the quantitative analysis of a series of parameters as lysozyme content, specific amino acid and mineral profiles and isotopes analysis (validation of the method in [7], or [8,9] and subsequent amendments thereof; see also [6]).

The combined use of analytical data and chemometrics in the traceability and authenticity of cheeses has been subjected to extensive investigation so far. This type of approach is defined as untargeted, in contraposition to targeted: the targeted or quantitative approach includes the identification of signals and assigning them to specific compounds, usually decided “a priori”, before the analysis. The untargeted or profiling approach consists of analyzing the whole spectrum of the sample and interpreting it as a unique fingerprint. A recent paper [10] discriminated pasture-based PDO Asiago cheese from hay-based milk with a metabolomic approach using NMR spectroscopy: 13 compounds were selected as the most discriminant ones with the aid of canonical discriminant analysis (CDA), but these results were found to be completely reliable only for short-ripened cheeses. PDO Asiago cheese [11] was studied using NMR spectroscopy to distinguish between cheeses produced in alpine farms from those produced in lowland farms or industries: many differences between the two groups were detected in the fatty acids composition. The geographical origin of buffalo milk and mozzarella cheese from two different areas (Campania and northern Apulia) was investigated using different analytical techniques in a very articulated study [12], where good results from the combination of isotopic parameters and NMR data were obtained. Even the geographical characterization of Parmigiano Reggiano was investigated [13], and a very high percentage of correct classification was obtained with NMR spectroscopy and chemometric analysis. A preliminary study on the grated form was performed with ATR-FTIR coupled with LDA to discriminate Parmigiano Reggiano from non-PDO cheese [14]. Focusing on Grana Padano cheese, different studies have been completed to authenticate it against non-PDO cheeses. For example, ultra-high-performance liquid chromatography coupled to quadrupole time-of-flight mass spectrometer (UHPLC/QTOF-MS) was used with an untargeted metabolomic approach to discriminate PDO Grana Padano cheese from non-PDO “Grana-type” cheeses with the aid of a chemometric analysis [5]. Other studies [15,16] focused on the characterization of the amino acid profile of Grana Padano cheese with NMR spectroscopy and studied the “typicality” of the free amino acid pattern of grated Grana Padano cheese to improve the already existing controls for assessing its compliance with PDO characteristics [15].

Within this framework, the current study aims to discriminate PDO Grana Padano cheese from other Italian PDO hard cheeses, its main competitors and a group of soft cheeses which could be used to adulterate its shredded form [17] by using NMR spectroscopy and multivariate analysis. The novelty of this study is that thanks to the wide variety of non-GP samples, the proposed approach could be the basis for the development of a tool which could be adopted from controlling bodies. Due to the untargeted nature of the analytical essay, no important chemical constituents are in principle disregarded, either in the aqueous and lipid fraction, obtaining a high performance in the discrimination. A parallel targeted approach allowed also for pinpointing some of the most important analytes which were driving the discrimination. Moreover, even other aspect could be investigated thanks to the overall view of the sample given by the NMR analysis. Ultimately, a very good discrimination is performed with NMR and Random Forest analysis, seldomly used with NMR data, which does not overfit the dataset, but thanks to its structure, the level of accuracy is very high.

## 2. Materials and Methods

### 2.1. Samples

A total number of 144 samples of cheese from different years (from 2017 to 2021) have been used for NMR analysis, which was provided by the Consortium. Among them, 56 samples were PDO Grana Padano cheeses, 32 samples of its main Italian competitor and 56 were foreign competitor cheeses (Table 1). In this way, it was possible to gather as heterogeneous a dataset as possible.

### 2.2. Experimental Procedures

Both the aqueous and lipid fractions of cheeses were considered, and it followed the method reported in [18], which is briefly explained below. After grating the cheeses, they were freeze-dried, and then about 100 mg was weighed. For the aqueous extractions, 900 μL deionized H_2_O (18.2 MΩ∙cm, Milli-Q, Millipore, Bedford, MA, USA) and 100 μL D_2_O (D_2_O, 99.9% isotopic purity containing 0.03% 3-(Trimethylsilyl)propionic-2,2,3,3-d4 acid sodium salt or TMSP-D4) were added. The solution was firstly shaken for 20 min at 1000 RPM and then centrifuged for 15 min at 12,000 RPM (9677 RCF). For the lipid extraction, 900 μL CDCl_3_ were added to the weighted cheese and then only shaken for 20 min without the centrifugation step. After filtration with a 0.22 μm PVDF filter (Millipore, Merck KGaA, Darmstadt, Germany), 600 μL was transferred to the 5 mm Norell 509-UP NMR tube. NMR is a fast and reliable technique, and, in this work, it was adopted to see if in the future it could become an alternative to chromatography for authenticity purposes. The spectra were recorded on a Bruker Avance Neo spectrometer working at 600 MHz base frequency, equipped with a broadband 5 mm Z-gradient probe, and the acquisition and processing were completed using TopSpin 4.1.3 software in the automation mode with Icon NMR 5.2.3. The spectra of the aqueous extracts were recorded with the following parameters: *noesygppr1d* pulse program with water signal suppression, the size of the spectrum (sweep width, SW) was 13.50 ppm, time domain (TD) consisted of 65,536 (64 K) data points, number of scans (NS) was 128, relaxation delay (D1) was 10 s and receiver gain (RG) was fixed at 16. The spectra of the lipid extracts were recorded with 2 experiments, the first consisting of a *zg30* pulse program with SW of 11.90, TD of 65,536 data points, NS of 64, D1 of 10 s and RG fixed at 20.2. The second experiment carried out with the lipid extract consisted of a *noesygppr1d.wvm* pulse program with the same parameters as the previous one except for an NS of 128, D1 of 3 s and RG set at 101. Spectra were processed in the TopSpin software with the size of real spectrum (SI) set to 131,072 (128 K, 2 × TD) data points, and a *apk0.noe* phase correction program was applied automatically to each spectrum.

The identification of 24 compounds was performed manually, based on the literature data (structures confirmed with 2D experiments) and with the aid of AssureNMR software v. 2020.09.23, using HMDB [19] and BBIOREFCODE [20] databases. The quantification of compounds in aqueous fraction was performed with AssureNMR software through an external standard method using the so-called ERETIC technique (electronic reference to access in vivo concentrations), using the 2 mM sucrose reference solution in the H_2_O and D_2_O mixture (9:1 *v*/*v*) as an external standard; the value was subsequently corrected by the weight of the sample. The quantification of lipids was performed following the same procedure, but the identification was performed only manually, and the results were calibrated to the integral area of alpha carboxyl protons (2.31 ppm) of all fatty acids [21] in percentages. The repeatability was tested through the comparison of the results obtained from the periodical acquisition of two manufacturer standards with the same parameters of the experiment described above (Andersen et al., 2023 [18]). The reproducibility was tested through the periodic acquisition of several cheese samples and the comparison of the obtained spectra at different times.

### 2.3. Statistical Analysis

All spectra were divided into buckets of 0.04 ppm, and no pH adjustments were needed to align them except for the peak of histamine, which showed shifts but did not affect the statistical analysis. The regions of TSP and water for aqueous and of TMS and chloroform for lipid fraction were excluded from the binning process. The bucket table was obtained using AssureNMR (v. 2020.09.23 License No. 128-27499325 valid until 24 December 2034, Bruker BioSpin GmbH, Rheinstetten, Germany) and then analyzed using the open-source (GPL-2 and GPL-3 licenses) R programming language [22]. All the data management and plots were performed with tidyverse [23] and tidymodels [24] packages (also open-source codes under MIT license). For the analysis of the lipid fraction, the bucket tables for the two NMR experiments were merged.

A simple PCA was performed to visualize any intrinsic variation of the dataset and the possible clustering of the samples or identifying outliers [25]. All the variables were previously Pareto scaled, which confers partially the same weight to each variable and allows for not giving so much importance to the regions with noise. After that, we decided to apply a Random Forest classification approach [26]. Random Forest was chosen over the more common Partial Least Square Discriminant Analysis (PLS-DA), because PLS tends to overfit the data, whereas in this case, RF can build a reliable classification model thanks to its structure: an ensemble of trees which are able to “protect” the model from the individual errors. The dataset was divided into training and test set; the training set was 75% of the total and stratified for the variety class (Grana Padano vs. Competitor class). The number of nodes was tuned to 16 with a repeated (5 times) 5-fold cross-validation scheme with a fixed number of trees of 100. The previous scheme was repeated 500 times to obtain an esteem of the variability in the predictive power. As a further validation step, the entire pipeline was then applied 500 times on a dataset with randomly shuffled class labels (results in Appendix A and Section 3.1). Every time, the model performance was assessed by calculating the Receiver Operating Curves (ROCs) (pROC package [27]). In all cases, accuracy, and AUC (Area Under the Curve) were used to quantify the predictive capacity. The variable importance was assessed in terms of Gini impurity index [28]. The aggregated variable importance across the 500 models was assessed by calculating the median rank of the individual variables.

## 3. Results and Discussion

Figure 1 sets out the aqueous and lipid spectra of PDO Grana Padano cheese acquired under the conditions.

The aqueous ^1^H experiment spectra (Figure 1a) are dominated by the signals of organic acids, such as acetate and lactate, and amino acids, such as alanine and valine.

Figure 1b, instead, shows the ^1^H experiment for the lipid fraction, in which the major signals belong to the methylenic and methyl protons of fatty acids. To better perceive the influence of minor compounds, samples were acquired also with a multi-suppression of the main signals. As it is shown in the inset of the figure, the signals of bis-allylic protons and caproleic and rumenic acid are increased.

### 3.1. Aqueous Fraction

A PCA was performed on the 144 samples considering both the aqueous and lipid fractions to visualize the intrinsic variability of the datasets and possible outliers. Figure 2A reports the PCA score-plot for the first two components related to the aqueous fraction, which describe 62.5% of the variance.

The two groups (Grana Padano samples vs. foreign and Italian competitors) are not clearly separated, meaning that the separation between the groups is not responsible for the larger fraction of the variability in the dataset. The plot clearly shows the presence of outliers, all of which belong to the competitor group, and, by looking at the metadata, most of them are very different from Grana-type cheeses even from an organoleptic point of view, being classified as soft cheeses. In Appendix A, the PCA plot for the aqueous fraction is reported, highlighting the different metadata (year, type of cheese, geographical origin as macro-areas), which shows a wide distribution in the space of the samples and their overlapping, meaning that all the mentioned characteristics influence the profile. The presence of a larger variability in the “competitors” group is not unexpected, since the samples belong to different cheese classes and, also for non-PDO hard cheeses, there is not a strict protocol on their production process. To obtain a better separation, we switched to a supervised classification approach comparing the performance of Random Forest and PLS-DA. Our results indicate that with the data at hand, RF classification outperformed PLS-DA (see Appendix A), so in the following, we will illustrate the results obtained by Random Forest.

Appendix A shows the ROCs curves for two different scenarios: in red, the curves built on the 500 models, and in gray, the curves built for the 500 models with shuffled class labels. The plot clearly indicates that the models with correct labels have better performances than the ones obtained with shuffled class labels.

The results in terms of AUC values for the aqueous fraction are shown in Figure 3A. In keeping to what can be observed for the ROC curves, the AUCs of the 500 models are close to 1 (0.93 for aqueous; 0.91 for lipid; 0.94 for unified), indicating an overall good predictive potential of the model. As a further confirmation of the goodness of the proposed approach, the AUCs of the models with permuted labels groups are around 0.5, which corresponds to the performance of a random classifier with a balanced dataset. A good feature of Random Forest is that a measure of the importance of the individual variables can be extracted from each individual model. In this case, it was possible to annotate some of the most relevant buckets and assign them to methyl and methylenic groups, which are typical of amino acids and lactic acid. These results suggested a prominent role of these metabolites in the discrimination between the two groups of samples. To confirm this evidence, the important buckets were annotated, and the raw NMR data were used to quantify lactic acid, tyrosine, proline, glycerol, valine, serine, isoleucine, and lysine, and the results are shown in Appendix A. The violin plot was chosen to represent the situation, because it can show the amounts in the various cheeses, and the heterogeneity of the dataset is still represented. The plot clearly indicates that the amount of lactate and tyrosine was higher in Grana Padano samples, whereas the amount of glycerol, proline, isoleucine, lysine, serine, valine, and alanine recorded in the Grana Padano samples falls within the wide range of competitor samples. These differences could be due to the diversity of amino acids required by various lactic acid bacteria used to produce cheeses [29]. Another important aspect that is highlighted looking at the violin plots is the strong homogeneity of the Grana Padano class compared to competitors: as already mentioned, this is not unexpected due to the large heterogeneity of the “competitors”. This somehow expected characteristic highlights how challenging it will be to define a threshold for authenticity. It is worth mentioning that the most extreme samples visible in the quantification plots belong to cheeses that are completely different from Grana Padano cheese or to some Grana Padano heading from a production starting from organic milk. These differences could be recognized merely by taste and sight when in slices form, and even from the NMR spectra, the differentiation among samples is evident. Interestingly, even if the quantified compounds contribute to the most discriminating buckets in the untargeted NMR analysis, a Random Forest classifier which considered only the intensity of the quantified compounds (targeted) showed a lower performance in comparison to the fully untargeted approach (Figure 3A). In this case, the median AUC was slightly lower even if clearly better than random. This result highlights an important strength of the untargeted approach over a targeted investigation.

### 3.2. Lipid Fraction

The lipidic fraction was treated as the aqueous one: the results of PCA are shown in Figure 2B, and the PCA plot highlighting the other metadata is shown in the Appendix A. The score-plot shows that the first two components explain the 48% of the variance, and no clear separation between the two classes of samples is visible. Also in this case, to improve the separation, the RF classification was implemented and the results in terms of AUCs are summarized in Figure 3B. Also here, the classification performance is close to 1, while its random counterparts center around the 0.5 chance value. In absolute terms, the lipid fraction seems to show a slightly lower performance compared to the aqueous one (Figure 4). As before, highly discriminating buckets can be putatively attributed to some lipid categories, even if a definitive assignment is not possible. Among unsaturated lipids, minor important signals can be assigned to linoleic, linolenic and rumenic acid. For the quantification of lipids, different methods were adopted, and the total amounts are expressed in % referring to the -CH_2_-COOH protons (2.31 ppm). The results are reported in Appendix A: linoleic acid and linolenic acid were quantified as suggested by [30], rumenic acid was quantified as suggested by [31] and triglycerides were quantified as suggested by [21]. As we can see, authentic Grana Padano samples show, in general, higher levels of linoleic acid, and this observation is particularly relevant due to the claimed health benefits of this compound [32]. Rumenic acid and linolenic acid are lower in PDO Grana Padano cheese: rumenic acid is expected to be in lower amounts because it comes from the biohydrogenation of CLA (conjugated linolenic acids), which are generally higher. In fact, the CLA amount depends on different factors, such as the cow’s diet, breed, and stage of lactation [33]. Differences in the lipid profile could be traced to the treatments that milk underwent before the production [5]. In keeping with what was observed for the aqueous fraction, violin plots show the heterogeneity of the competitor class compared to the authentic class.

As before, a RF classifier was then applied only to the most important quantified features. It showed a lower performance in comparison to the fully untargeted one (Figure 3B), reinforcing the idea that a full spectrum is more informative than a subset of well-characterized peaks, even if in the absence of a clear-cut annotation it is impossible to understand which are the major chemical drivers of the differentiation, highlighting the fact that a combination of the untargeted and targeted approaches has advantages.

To push forward this observation, we investigated the potential of a unified untargeted approach merging the bucket tables obtained in the analysis of the aqueous and lipid fractions. The results of the RF classifier are summarized in Figure 3C. As before, the AUC of the classifier with correct labels is close to 1, and it shows a marginal, but clear, improvement in comparison to what was obtained using the separate fractions (Figure 4). In terms of interpretation, the median value of the AUCs for the unified model is 0.95 with an interquartile range of variability of 0.05; for the aqueous model, it is 0.93 with an interquartile range of 0.06 and for the lipid model, it is 0.92 with an interquartile range of 0.06. The fact that the ability to correctly classify samples increases with the “fusion” of the aqueous and lipid fractions suggests that their information content is somehow complementary. Amino acids are indeed up-regulated and determined by the cows feeding, so they are key factors for the quality of cheese, whereas lipids are down-regulated depending on the processing specifications: possible counterfeiting can yield differences in this part [5]. By delving deeper into the misclassifications, what emerges is that four samples were never correctly classified: two are real PDO Grana Padano samples, one is a competitor sample, and the last one is a Grana Padano, but the cows were only fed with hay, so some kind of differentiations from the traditional ones are expected.

As reported in the introduction section, the authentication issue is becoming a central topic in the analytical world. To date, the preferred analytical techniques are chromatography and IR spectroscopy. Liquid chromatography coupled with mass spectrometry was adopted on Parmigiano Reggiano cheese [34] and on GP cheese [5]. In both works, there is a long paragraph describing the method, because an untargeted approach with chromatography is hard to apply mainly due to the annotation and calibration of signals. IR spectroscopy was adopted on Parmigiano Reggiano cheese [35] also in combination with ATR [14], but the main disadvantage of this method is the lack of signal assignation to specific compounds. Chemometric analysis was combined with both the above-mentioned techniques, being a powerful tool to deal with the large amount of data obtained. Actually, our approach combines the advantages of both the techniques: the compounds identification of chromatography, being the results of the cited literature in accordance with the results of this work, and the easy untargeted approach of IR spectroscopy to accomplish the fixed task.

## 4. Conclusions

PDO Grana Padano cheese authentication is one of the main topics for the Consortium [36], which oversees its protection and promotion to consumers. NMR spectroscopy is a powerful technique in this field, because with just a small amount of sample and a simple and reproducible analysis, can provide important evidence about the authenticity of Grana Padano samples. The workflow is summarized in Figure 5.

It is important to notice that even if it is possible to build two separate models for the two fractions, the aqueous and the lipid one, the most reliable results are obtained when they are merged, meaning that it is important to look at the sample in its entirety. That is why the untargeted approaches are becoming even more important in recent years. This result could pave the way to new studies on the discrimination of cheeses through an analysis of the two fractions considering the entire fingerprint, as it was completed in this work. In this way, even more “marginal” information as the month or the region of production can be investigated having a holistic view of the samples. Our work clearly shows that a Random Forest classifier could represent a powerful tool to discriminate between “original” and “competitor” samples; however, it is important to point out that the reliability of such an approach increases with the size and the representativeness of the dataset. Anyway, at these early stages of development of this approach, it could be used to support the results of the official methods based on the isotopic analysis: in the case of an “undefined” samples, our method could be used as a clarifier.

In conclusion, it is fair to say that NMR is an expensive analytical tool and not many laboratories have access to this machinery, but the advantages of short and simple analysis are undisputable. Moreover, benchtop NMR devices having lower prices are appearing on the market, and they are already being used in foodomics [37], in particular dairy analysis [38], and they could be adopted for such analysis after an improvement of this exploratory model.

## Figures and Tables

**Figure 1 foods-13-00358-f001:**
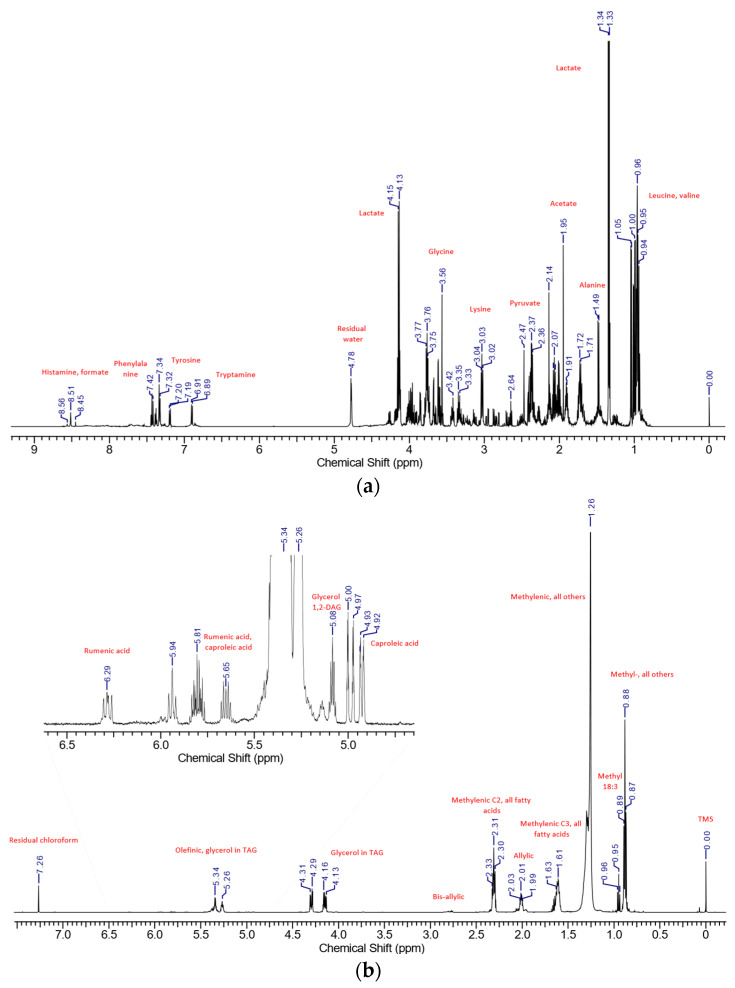
(**a**). The ^1^H NMR spectrum of the aqueous extract of PDO Grana Padano cheese: signals with assignments are tagged. (**b**). The ^1^H NMR spectrum of the lipid extract of PDO Grana Padano cheese: obtained with zg30 pulse program experiment: signals with assignments are tagged. The zoom above is showing low but important signals, which are highlighted through the acquisition of samples using the *noesygppr1d.wvm* pulse program experiment.

**Figure 2 foods-13-00358-f002:**
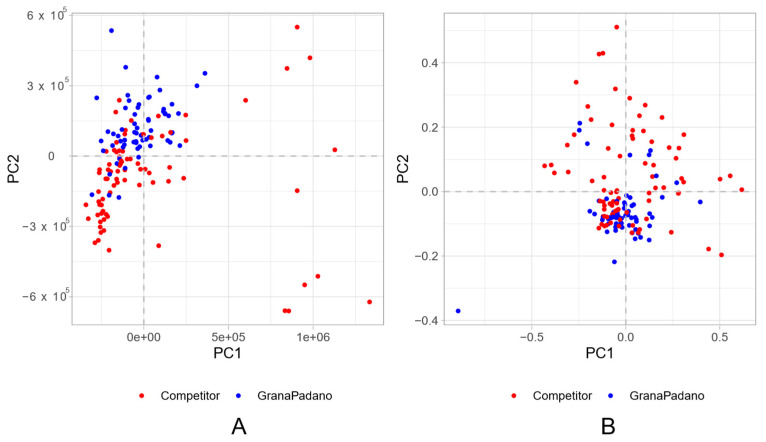
(**A**) Bidimensional PCA plot for the aqueous fraction of samples describing 62.5% of the variance; (**B**) bidimensional PCA plot for the lipid fraction of samples describing 48% of the variance. The dots are colored according to their variety, PDO Grana Padano cheese or competitor.

**Figure 3 foods-13-00358-f003:**
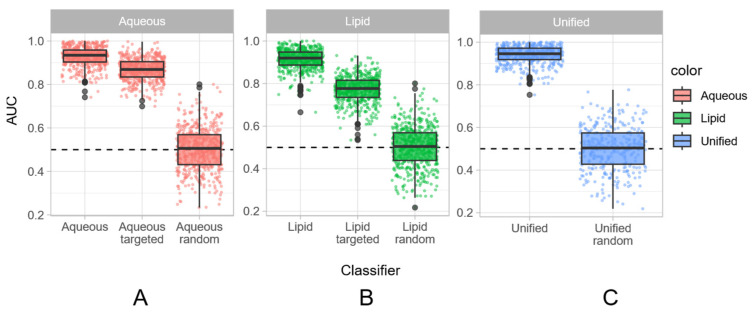
Boxplot of AUC values for the various models described in the paper. In each plot, the untargeted model shows the highest values of AUC, meaning that it has better performances than the targeted one. (**A**) Aqueous model; (**B**) lipid model; (**C**) unified model.

**Figure 4 foods-13-00358-f004:**
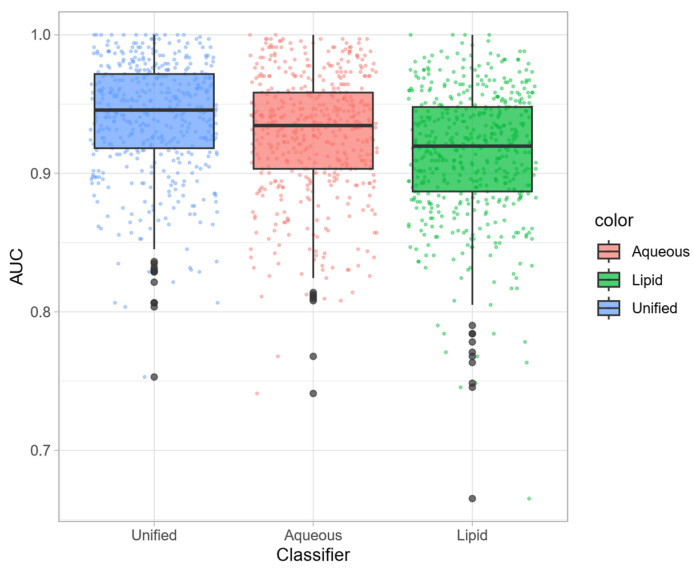
Boxplot of AUC values for the untargeted aqueous, lipid and unified models. The mean value of AUCs of the unified model is the highest among the three models, highlighting the better performances.

**Figure 5 foods-13-00358-f005:**
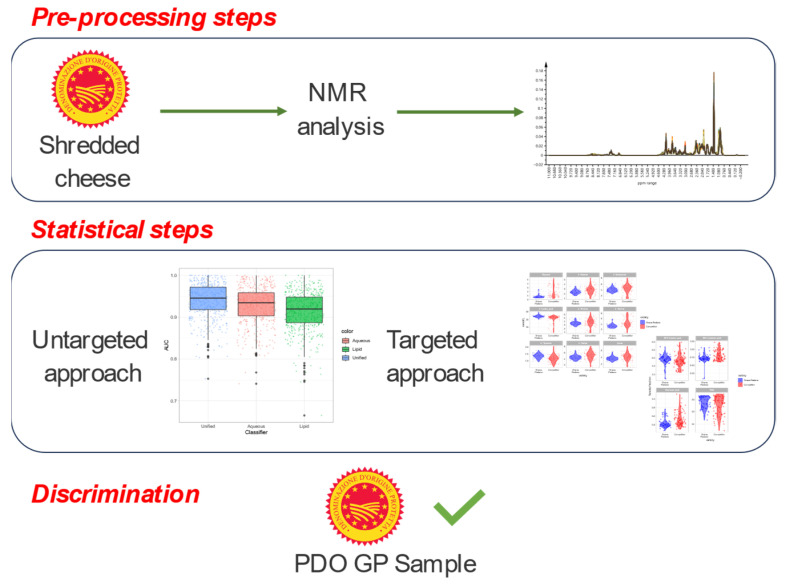
Workflow with the important steps described in this paper.

**Table 1 foods-13-00358-t001:** Summary of the samples, reporting their number according to the type, the geographical origin and the year of production.

Cheese Type	Geographical Origin	Year	No. of Samples
PDO Grana Padano	North of Italy	2017–2018–2019–2020	57
Grana Padano cheese with feed modifications	North of Italy	2021	7
PDO Italian cheese	North of Italy	2017–2019–2020	32
Italian hard cheese	North of Italy	2017–2018–2020–2021	14
“Parmesan”	USA	2017	4
Hard cheese	Czech Republic	2017–2018–2020–2021	8
Hard cheese	Lithuania	2017	1
Hard cheese	Estonia	2017	1
Hard cheese	Austria	2018–2020–2021	5
Soft cheese	Belgium	2018	1
Soft cheese	Holland	2018–2020	2
Hard cheese	Poland	2018–2020–2021	4
Hard cheese	Hungary	2020	1
Soft cheese	Switzerland	2020	1
Soft cheese	France	2020	3
Soft cheese	Czech Republic	2020	1
Soft cheese	Germany	2020	2

## Data Availability

The data presented in this study are available on request from the corresponding author. The data are not publicly available due to confidentiality.

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
