# Peer review of "1H-NMR Approach for the Discrimination of PDO Grana Padano Cheese from Non-PDO Cheeses"

_foods, 2024, doi:10.3390/foods13030358_

Round 1

Reviewer 1 Report

Comments and Suggestions for Authors

Maestrello et al. have investigated on the application of 1H-NMR metabolomic approach combined with multivariate analysis to build a classification model capable of discriminating PDO Grana Padano cheese from a large dataset of hard-cheese competitors. It is an interesting piece of work and should be of interest to food scientists and nutritionists. However, the following issues should be addressed:

1.        Although the authors claim the application 1H-NMR approach for discrimination as an easy and viable method, the accessibility to this sophisticated and expensive instrument available in only certain major research centers can reduce the application of this method. The authors should comment on this issue, that is, on (1) accessibility and (2) cost per sample to use 1H-NMR.

2.        The abstract is very plain and descriptive. It should be provided with technical approach, qualitative and quantitative information on using 1H-NMR for discrimination of PDO Grana Padano cheese and non-PDO hard cheeses.

3.        A clear-cut, straightforward objective of this study should be provided at the end of introduction section.

4.        Why sections under section 2 are labelled as 2.1?

5.        The long single paragraph in section “experimental procedures” should split into two paragraphs; probably starting “The second experiments consisted….” as second paragraph.

6.        The statistical software used for statistical analysis along with its purchase details should be provided.

7.        The key groupings of data in Figure 2 should marked directly inside the PCA plot and shown.

8.        A schematic diagram showing the 1H-NMR approach for discrimination along with the qualitative, quantitative and validation parameters obtained as outcomes should be provided. This would enable the readers with an easy take away points at a glance.

9.        A comparative discussion on the discrimination approach by 1H-NMR reported in this study with that by other reported techniques would highlight the superiority of 1H-NMR over other reported ones.

Comments on the Quality of English Language

Minor editing of English language required

Author Response

Reviewer 1:

Comments and Suggestions for Authors

Maestrello et al. have investigated on the application of 1H-NMR metabolomic approach combined with multivariate analysis to build a classification model capable of discriminating PDO Grana Padano cheese from a large dataset of hard-cheese competitors. It is an interesting piece of work and should be of interest to food scientists and nutritionists. However, the following issues should be addressed:

  1. Although the authors claim the application 1H-NMR approach for discrimination as an easy and viable method, the accessibility to this sophisticated and expensive instrument available in only certain major research centers can reduce the application of this method. The authors should comment on this issue, that is, on (1) accessibility and (2) cost per sample to use 1H-NMR.

We added a comment on this disadvantage of NMR at lines 363 to 367.

  1. The abstract is very plain and descriptive. It should be provided with technical approach, qualitative and quantitative information on using 1H-NMR for discrimination of PDO Grana Padano cheese and non-PDO hard cheeses.

Thank you for your suggestion, which could make this paper more appealing to other scientists. We added what was required in lines 18 to 22.

  1. A clear-cut, straightforward objective of this study should be provided at the end of introduction section.

We clarified the objective at lines 82 to 86 and 88 to 89.

  1. Why sections under section 2 are labelled as 2.1?

We corrected the numbering of the subparagraphs

  1. The long single paragraph in section “experimental procedures” should split into two paragraphs; probably starting “The second experiments consisted….” as second paragraph.

We decided to split the paragraph at line 133, separating in this way the “operative” NMR part from the post-acquisition operations.

  1. The statistical software used for statistical analysis along with its purchase details should be provided.

We specified the details at 151 to 157.

  1. The key groupings of data in Figure 2 should marked directly inside the PCA plot and shown.

We inserted in the Supplementary Material the PCA plots for aqueous and lipid fractions with the key groupings highlighted by the different colors (Figure A.4). We decided to make three different plots otherwise it could be confusing.

  1. A schematic diagram showing the 1H-NMR approach for discrimination along with the qualitative, quantitative and validation parameters obtained as outcomes should be provided. This would enable the readers with an easy take away points at a glance.

Thank you very much for your helpful suggestion. We inserted it as Figure 5, lines 344 to 347.

  1. A comparative discussion on the discrimination approach by 1H-NMR reported in this study with that by other reported techniques would highlight the superiority of 1H-NMR over other reported ones.

Thank you for the suggestion, we inserted a short discussion at lines 325 to 338.

Comments on the Quality of English Language

Minor editing of English language required.

The text was already checked by an English mother tongue service.

Reviewer 2 Report

Comments and Suggestions for Authors

The manuscript of Maestrello et al. describes 1H NMR – based discrimination of PDO Grana Padano cheese from non-PDO cheeses. Altogether 144 samples were analyzed as a water and chloroform extracts. It is said that samples originate from three groups, however, they were ultimately merged only into two groups which were subjected to final discrimination. In my opinion, the manuscript is written in a very vague form neglecting many of important details. The main finding that the fingerprinting data outperform those obtained by compound profiling is valid only when there is not enough compound identified, otherwise it is not correct. I believe, this finding is obvious and therefore also well known. The manuscript is of potential interest, however there are several important issues that have to be solved. In its current form, the manuscript does not reach the standards of Foods.

Major points:

-        The authors do not let the reader to touch the samples. There is no list of sampled cheese. It would be beneficial to see the details like at least cheese type and region of production.   

-        It seems that there are three groups of samples, but two groups were merged into one. It would be interesting to investigate also e.g. discrimination between PDO and Italian no-PDO and also Italian non-PDO to foreign non-PDO.

-        It would be great to perform also the mentioned PLS analysis and compare the results with those provided by RF. At least self-explaining PLS score plots could be then shown.

-        PCA also indicates there are some outlying samples. What about their exclusion or at least some discussion there? Again there should be a list of samples analyzed or subdivision into three groups.

-        The only information on the RF performance values can be derived only from figures 3 and 4.

-        There should be a regular SI file not three pictures without any caption. The SI pictures are presented in the main text in a superfluous detail which is confusing and mistakable with the discussion devoted to regular pictures. Fig A1 should be remove as it has no value for the manuscript. Figs A2 and A3 should show also the median values otherwise it has a limited value too.

-        The terms untargeted and targeted analysis should be defined. Terms fingerprinting/binning/bucketing vs compound profiling/concentration data etc. should be used because for compound profiling we can talk also about targeted and untargeted analysis.  

Minor points:

-        P2 row 63. UHPLC should be called ultra-high-performance liquid chromatography

-        P3 row 108. For zg30 use italic

-        P3 row 109. It is not clear to what pulse sequence the lipid noesy is linked; to the other lipid measurement (probably yes, when later we learn that both experiments were merged) or to the other noesy experiment.

-        P3 row 124. Periodical acqusition

-        The information how many compounds were identified/quantified would be handy.

-        Were there some excluded regions from 1H NMR spectrum before bucketing?

-        Fig. 1b. What lipid experiment is it; zg30 or noesy?

-        Fig 4’s caption should be more descriptive as its the relevant discussion is 2 pages away in the main text.

-        P8 row 383. What is Consortium?

I believe that all the points mentioned are relevant and should have been considered prior to submission.

Comments on the Quality of English Language

there are some minor issues.

Author Response

Reviewer 2:

Comments and Suggestions for Authors

The manuscript of Maestrello et al. describes 1H NMR – based discrimination of PDO Grana Padano cheese from non-PDO cheeses. Altogether 144 samples were analyzed as a water and chloroform extracts. It is said that samples originate from three groups, however, they were ultimately merged only into two groups which were subjected to final discrimination. In my opinion, the manuscript is written in a very vague form neglecting many of important details. The main finding that the fingerprinting data outperform those obtained by compound profiling is valid only when there is not enough compound identified, otherwise it is not correct. I believe, this finding is obvious and therefore also well known. The manuscript is of potential interest, however there are several important issues that have to be solved. In its current form, the manuscript does not reach the standards of Foods.

Major points:

1- The authors do not let the reader to touch the samples. There is no list of sampled cheese. It would be beneficial to see the details like at least cheese type and region of production.

Thank you for your helpful suggestion, we inserted a summary of the analyzed samples in Table 1, lines 102 to 104

2- It seems that there are three groups of samples, but two groups were merged into one. It would be interesting to investigate also e.g. discrimination between PDO and Italian no-PDO and also Italian non-PDO to foreign non-PDO.

Thank you for your suggestion. Unfortunately, the “third” group formed by Italian PDO samples, a competitor of Grana Padano cheese, and the differences between the two PDO cheeses are already well-known, so we decided not to mention this. Moreover, we decided to focus only on Grana Padano cheese and discriminate it from “all the rest”, from everything else. We can dig into those differences in the next studies.

3- It would be great to perform also the mentioned PLS analysis and compare the results with those provided by RF. At least self-explaining PLS score plots could be then shown.

As you suggested, we enlarged the discussion part with the PLSDA model at line 317 to 324.

4- PCA also indicates there are some outlying samples. What about their exclusion or at least some discussion there? Again there should be a list of samples analyzed or subdivision into three groups.

The outliers are mostly assigned to soft-cheeses, underlining the complete difference of NMR profile. We decided to keep them into the analysis because we wanted a model which is reliable with a large variety of different cases. It could happen that if not trained well enough, the model could assign as real GP a soft-cheese because it has “never seen” a sample like that, whereas through the inclusion of those outliers we trained it as much as we could. We inserted the specification about this at line 218 to 222.

5- The only information on the RF performance values can be derived only from figures 3 and 4.

Unfortunately, RF does not give pictures to visualize the model and the Figures (3 and 4) can summarize correctly the performances of the models. In Supplementary material Figure A1 is important for the model performances, because it highlights the AUC values of 500 RF model and 500 RF models with random labels. To clarify the differences among the models we inserted the medians and interquartile ranges of the AUCs of each model (line 305-307).

6- There should be a regular SI file not three pictures without any caption. The SI pictures are presented in the main text in a superfluous detail which is confusing and mistakable with the discussion devoted to regular pictures. Fig A1 should be remove as it has no value for the manuscript. Figs A2 and A3 should show also the median values otherwise it has a limited value too.

We proceeded with the SI file and we improved the discussion of the pictures. We also improved the captions of the figures, but we think that Fig. A.1 is important for the validation of the model as reported in comment 5. The median was added, even though the important thing is the visualization of the distribution of data, not just a marker value.

7- The terms untargeted and targeted analysis should be defined. Terms fingerprinting/binning/bucketing vs compound profiling/concentration data etc. should be used because for compound profiling we can talk also about targeted and untargeted analysis.

Thank you for the suggestion to clarify those concepts. We added the related description at line 50-56.

Minor points:

- P2 row 63. UHPLC should be called ultra-high-performance liquid chromatography

We corrected it, new line 72

- P3 row 108. For zg30 use italic

We corrected it, new line 126

- P3 row 109. It is not clear to what pulse sequence the lipid noesy is linked; to the other lipid measurement (probably yes, when later we learn that both experiments were merged) or to the other noesy experiment.

We have explained this better in the text, new line 127

- P3 row 124. Periodical acquisition

We corrected it, new line 143 to 144

- The information how many compounds were identified/quantified would be handy.

The compounds were identified and quantified on the base of the important buckets; thus, we can add just the number of identified compounds at the very beginning of our study while we were trying to understand the signals of the spectra. Line 133

- Were there some excluded regions from 1H NMR spectrum before bucketing?

Yes, added at line 151 to 152

- Fig. 1b. What lipid experiment is it; zg30 or noesy?

We clarified it in the caption of the figure, new lines 191 to 194.

- Fig 4’s caption should be more descriptive as its the relevant discussion is 2 pages away in the main text.

We moved the figure to the new lines 295 to 299, and modified captions make it clearer.

- P8 row 383. What is Consortium?

We added a new reference No. 35 to clarify it (line 341).

I believe that all the points mentioned are relevant and should have been considered prior to submission.

Comments on the Quality of English Language

there are some minor issues.

The text was already checked by an English mother tongue service.

Round 2

Reviewer 1 Report

Comments and Suggestions for Authors

The authors have satisfactorily addressed all the comments raised by reviewers.

Comments on the Quality of English Language

Minor editing of English language required

Author Response

The text was revised by a native English speaker employed by the company EUROSTREET SOCIETÀ COOPERATIVA, Italy, https://www.eurostreet.it/

Reviewer 2 Report

Comments and Suggestions for Authors

The manuscript did not improve much with the revision. When I see the “care” spent on the revision I makes me doubt about the whole content. It only underlines the impression I got when I read the article for the first time.

The new supporting materials do not correspond to the main text. There is different order of the figures than in the referencing paragraph. In figure A1, there is color coding referring to the cheese type; there are only 7 PD??? The violin plots should show also the median values not only the diversity of the samples, that is just one mark, but it shows a lot. The text refers to PSL-DA, but SI contains PCA according to the caption. I still do not understand why a single file cannot be made.

When unsupervised PCA indicates there are some outlying samples, they should be removed otherwise even supervised statistical methods cannot help. I still believe that PLS would discriminate the two groups when soft cheese is removed. The response that this group has to be part of the data set is in my opinion false.

Definitely, the authors did not utilize the potential what the obtained data have.

The added sentence into conclusion about benchtop NMR is rather funny as for foodomics at least 400 MHz spectrometers are used and benchtops do not reach even half of this frequency.

Author Response

The manuscript did not improve much with the revision. When I see the “care” spent on the revision I makes me doubt about the whole content. It only underlines the impression I got when I read the article for the first time.

We apologize for the inaccuracies present in the first revision and in this second run of revisions we did our best to clarify and re-organize our manuscript. In particular:

  • In the introduction we better clarified the aim of the study explaining why we decided to include a large diversity of samples in our dataset (lines 81 to 88);
  • We re-organized and better documented the supplementary material;
  • We included a more detailed discussion of the PLS-DA results in a specific section of the Supplementary Material, mentioning the application of this data analysis approach in the main text (lines 219 to 224).

The new supporting materials do not correspond to the main text. There is different order of the figures than in the referencing paragraph.

In figure A1, there is color coding referring to the cheese type; there are only 7 PD???

Figure A1 (renamed as S1, as the guidelines ask) was used to highlight the additional characteristics of our samples which were not visible in the plot in the main text (Figure 2). In particular, the first figure highlights three different classes of samples: 1) “hard” cheese (including PDO Grana Padano), 2) Grana Padano w/ organic feed (real Grana Padano produced from milk from organic management). 3) Soft cheese which were included in the dataset due to their potential use in adulteration of shredded Grana Padano. We inserted an explanation in the figure caption.

The violin plots should show also the median values not only the diversity of the samples, that is just one mark, but it shows a lot.

The median value was added to the plots as suggested by the reviewer.

The text refers to PSL-DA, but SI contains PCA according to the caption. I still do not understand why a single file cannot be made.

We apologize for the mislabeling. The PLS-DA score plot are now included in the Supplementary Material as Figure S6. All the supplementary material was combined in a unique file as suggested.

When unsupervised PCA indicates there are some outlying samples, they should be removed otherwise even supervised statistical methods cannot help. I still believe that PLS would discriminate the two groups when soft cheese is removed. The response that this group has to be part of the data set is in my opinion false.

We agree on the fact that with the elimination of outliers and the soft cheeses most likely the PLS would be able to classify the samples. But the objective of our investigation was to assess the potential of NMR in discriminating PDO from a representative group of cheeses which can be used to adulterate PDO Grana Padano in particular in its shredded form. For this reason, as we discussed in the introduction soft cheeses were included (DOI: 10.1111/1471-0307.12818) and these are exactly those samples which show up as outliers in the dataset, in particular in the aqueous fraction (see the PCA in Figure S1). The rationale behind this choice was to present to the classification algorithm a dataset which was as much as possible representative of the diversity which could be found in a “real life” scenario. It is indeed known that an efficient classifier could show a high misclassification rate on "never seen" samples or classes of samples. Most likely, the large diversity of our dataset is the reason why a more flexible classifier (random forest) worked better than PLS-DA in our specific scenario. Since we agree with the reviewer that PLS-DA is an approach highly popular in chemometrics, we present the results obtained with this model on our dataset in the Supplementary Material

The added sentence into conclusion about benchtop NMR is rather funny as for foodomics at least 400 MHz spectrometers are used and benchtops do not reach even half of this frequency.

While it is correct that benchtop NMR devices do not reach even half of the base frequency of the spectrometers usually employed in food analysis (80 MHz vs. 400 MHz), there have been already numerous studies employing these low-field machines in foodomics, such as those summarized in the review (https://doi.org/10.1016/j.aca.2023.341495). The final sentence in the conclusion has been corrected to include this reference (lines 361 to 364).